# Trust in and Use of COVID-19 Information Sources Differs by Health Literacy among College Students

**DOI:** 10.3390/healthcare11060831

**Published:** 2023-03-11

**Authors:** Xuewei Chen, Darcy Jones McMaughan, Ming Li, Gary L. Kreps, Jati Ariati, Ho Han, Kelley E. Rhoads, Carlos C. Mahaffey, Bridget M. Miller

**Affiliations:** 1School of Community Health Sciences, Counseling and Counseling Psychology, Oklahoma State University, Stillwater, OK 74078, USA; darcy.mcmaughan@okstate.edu (D.J.M.); hohan@okstate.edu (H.H.); kelley.rhoads@okstate.edu (K.E.R.); 2Department of Health Sciences, Towson University, Towson, MD 21252, USA; mli@towson.edu; 3Center for Health and Risk Communication, George Mason University, Fairfax, VA 22030, USA; gkreps@gmu.edu; 4School of Educational Foundation, Leadership, and Aviation, Oklahoma State University, Stillwater, OK 74078, USA; jariati@okstate.edu; 5Department of Psychology, Universitas Diponegoro, Semarang 50275, Indonesia; 6College of Health and Human Sciences, Purdue University, West Lafayette, IN 47906, USA; cmahaffe@purdue.edu; 7Arnold School of Public Health, University of South Carolina, Columbia, SC 29208, USA; bridget_miller@sc.edu

**Keywords:** health information trust, health information use, COVID-19 information sources, health literacy

## Abstract

People’s health information-seeking behaviors differ by their health literacy levels. This study assessed the relationship between health literacy and college students’ levels of trust in and use of a range of health information sources of COVID-19. We collected data from August to December 2020 among college students (*n* = 763) through an online survey. We used a health literacy measure containing three self-reported survey questions, developed by the CDC. We assessed the extent to which participants trusted and used any of the sixteen different sources of information about COVID-19. Respondents reported high levels of trusting and using COVID-19 information from the CDC, health care providers, the WHO, state/county/city health departments, and official government websites when compared to other sources. After controlling for demographic characteristics (i.e., gender, age, race, ethnicity, and income), those who reported having lower health literacy were significantly less likely to trust and use COVID-19 information from these health authorities when compared to participants who reported having higher health literacy. Students with lower self-reported health literacy indicated not trusting or using official health authority sources for COVID-19 information. Relying on low-quality information sources could create and reinforce people’s misperceptions regarding the virus, leading to low compliance with COVID-19-related public health measures and poor health outcomes.

## 1. Introduction

The regular and comprehensive communication of public health information is an important step in containing the spread of COVID-19, protecting people at risk of serious complications or death, and reducing the burden on the health system [1]. Diffusion of knowledge, especially about the COVID-19 virus, is also critical to promoting social, political, and economic development [2]. The COVID-19 pandemic created an *infodemic* [3], where a vast amount of information, including misinformation and disinformation, spread rapidly and impeded effective crisis management [4,5]. Misinformation refers to false information that is created and spread, regardless of an intent to harm or deceive, and disinformation refers to false information intended to be deliberately deceptive [6]. The vast amounts of COVID-19 information led to information overload, which includes the presence of unpleasant emotions (e.g., feeling overwhelmed), a reluctance to follow suggested health behaviors (e.g., physical distancing and wearing a mask), and information avoidance [5,7]. For example, when people experience emotional strain (e.g., confusion, frustration, fear) by what they characterize as excessive, inconsistent, conflictive, or inaccurate COVID-19 information, they may cope by setting boundaries, limiting the amount of information they are exposed to (i.e., not seeking information every day and not reading every piece of information), and limiting the sources of information they attend to [8].

Individuals with lower health literacy face greater challenges in evaluating the quality of health information and differentiating misinformation/disinformation from accurate information [6,9]. According to Healthy People 2030, personal health literacy is “the degree to which individuals have the ability to find, understand, and use information and services to inform health-related decisions and actions for themselves and others” [10,11]. Those with low levels of health literacy may hold misconceptions, such as assuming, incorrectly, that farmers have strong immune systems, with low chances of contracting COVID-19; these misconceptions might lead to a lack of preventative behaviors that protect against the virus [8]. A recent study conducted among undergraduate students majoring in healthcare in South Korea reported that e-health literacy (skills in obtaining and comprehending online health information) was positively associated with COVID-19 preventive behaviors [12]. Therefore, identifying the sources that are preferred by individuals with low health literacy for health information related to COVID-19 can provide evidence-based guidance for public health professionals, in order to identify the best sources for disseminating high quality and easy-to-understand COVID-19 information to different populations. These sources can be used to reduce uncertainty, decrease knowledge gaps, and, therefore, diminish health disparities among individuals with low health literacy.

People’s health information-seeking behaviors differ by their health literacy levels [6,9]. A previous study found that people with low health literacy are less likely to use and trust general health information from sources of authority (e.g., medical websites and health professionals), but are more likely to use and trust health information from social media (e.g., YouTube and celebrity blogs), which often contains inaccurate information [13]. However, individuals may use different strategies to seek information, depending on the specific health topics [14]. For example, the Health Information National Trends Survey (HINTS) data have shown that people in the U.S. seek cancer-related health information primarily from the Internet [15,16], people in Germany primarily seek cancer-related health information from their health care providers [17], and people in China primarily use the television to seek cancer information [18]. Another recent study found that people in New York City use a variety of sources to access information about dietary supplements, including the Internet, product packaging, books, friends, pharmacists, dietitians/nutritionists, and family members [19]. These findings suggest that focusing on specific groups of information seekers, health literacy levels, and predefined health topics will provide the opportunity for an in-depth analysis and comparison of health-information-seeking behaviors by a specific audience about COVID-19. Thus, we focused, in this study, on examining as to how U.S.-based college students with lower levels of health literacy seek health information about COVID-19. This study will contribute to the field because the literature shows that low health literacy skills can influence one’s exposure to misinformation and disinformation related to COVID-19, leading to information avoidance [8,9], which might accelerate uncertainty and create difficulty in making appropriate decisions about COVID-19-preventive behaviors [6].

Thus, the purpose of this study was to examine the relationship between health literacy and people’s trust, and to use of a range of potential health information sources for COVID-19 among college students. We proposed the following research questions:

What sources were highly trusted and frequently used by our participants for COVID-19 information?

How does health literacy play a role in our participants’ trust in and use of a range of potential health information sources for COVID-19?

## 2. Materials and Methods

### 2.1. Procedures and Participants

Data for the present analyses were derived from a larger cross-sectional online survey study, designed to investigate college students’ experiences during the COVID-19 pandemic [20]. Recruitment and data collection, using Qualtrics, were conducted from August 2020 to December 2020. The larger study was approved by Oklahoma State University Institutional Review Board. Detailed procedures are reported by McMaughan et al. [20]. Of the 849 students who submitted survey responses, 72 were dropped for straight line, missing, speedy or fake responses. An additional 14 were dropped due to failing the validation item associated with the trust in sources section, resulting in a final sample of 763 participants in our data analyses for this current study.

### 2.2. Measures

#### 2.2.1. Health Literacy

We used a health literacy measure developed by the Centers for Disease Control and Prevention (CDC) in the 2016 Behavioral Risk Factor Surveillance System (BRFSS) [21]. The measure contains three self-reported survey questions: (1) assessing individuals’ abilities to find information, “How difficult is it for you to get advice or information about health or medical topics if you needed it?”; (2) understanding oral information, “How difficult is it for you to understand information that doctors, nurses, and other health professionals tell you?”; (3) understanding written information, “You can find written information about health on the Internet, in newspapers and magazines, and in brochures in the doctor’s office and clinic. In general, how difficult is it for you to understand written health information?”. Each item was assessed on a 4-point Likert type scale ranging from 1 = “very difficult” to 4 = “very easy”. We calculated the sum score, with possible scores ranging from 3 to 12. A higher score indicated that the participant reported having lower health literacy. Furthermore, to identify the individuals who were at high risk of having a low health literacy, we defined low health literacy as a response of “very difficult” or “difficult” to at least one of these three questions [22]. In addition, the item assessing an individual’s ability to find information included a response of “I don’t look for health information”, and the item assessing the understanding of written information included a response of “I don’t pay attention to written health information”. Both responses were coded as 0.

#### 2.2.2. Information Trust and Use

We also assessed the extent to which participants trusted and used sixteen sources (see Table 1) of information about COVID-19.

#### 2.2.3. Demographics

Demographics included gender, age, ethnicity, race, and annual family income. Race and ethnicity were determined by asking participants to select from a list of racial (i.e., White, Black or African American, American Indian or Alaskan Native, Asian, and Native Hawaiian or Pacific Islander) and ethnic (i.e., Hispanic, Latinx, or of Spanish origin) categories, those which best matched their own identity. Multiple options could be selected. Gender included three options: male (including transgender men), female (including transgender women), and self-described (non-binary, gender-fluid, agender, etc.), with an option to provide an open answer. Participants were asked to identify their annual family income through the following options: less than $ (U.S. Dollar) 20,000, $20,000–$34,999, $35,000–$49,999, $50,000–$74,999, $75,000–$99,999, and over $100,000.

### 2.3. Data Analysis

Individuals who chose either “I don’t look for health information” or “I don’t pay attention to written health information” were treated separately from those who self-reported their levels of difficulty in finding information, understanding oral information, and understanding written information. We performed independent t-tests to compare trust in and use of different sources for COVID-19 information, between those who chose either “I don’t look for health information” or “I don’t pay attention to written health information”, and those who self-reported their levels of difficulty in finding information, understanding oral information, and understanding written information.

Among participants who indicated levels of difficulty in finding information, understanding oral information, and understanding written information, we investigated the relationship between their self-reported health literacy capacity (independent variable) and their trust in and use of each source of COVID-19 information (outcome variable), and we performed bivariate linear regressions (without covariates) and multiple linear regressions (controlling for demographic characteristics). We included the demographic variables (gender, age, ethnicity, race, and annual family income) in our multiple linear regression analysis as covariates, because these demographic variables are associated with health literacy based on prior research [23]. We conducted separate regressions for each COVID-19 health information source, in terms of participants’ trust in and use of them. We used Stata 16 for statistical analysis. The significance level was set at α = 0.05.

## 3. Results

Most participants were White (75%), women (66%), and between 18 and 24 years old (80%). Participants’ ages ranged from 18 to 59 (M = 23.13, SD = 7.38). Table 2 shows the demographic characteristics of our participants.

A total of 87 participants (11.40%) chose either “I don’t look for health information” (54 participants) or “I don’t pay attention to written health information” (48 participants).

Among those who indicated their levels of difficulty in finding information, understanding oral information, and understanding written information (*n* = 676, 88.60%), their health literacy scores ranged from 5 to 12 (M = 9.75, SD = 1.56). The health literacy score data were moderately skewed towards the high end, which indicated that most of our participants reported having adequate health literacy (*n* = 526, 68.94%). About 19.66% of our participants (*n* = 150) were at a high risk of having low health literacy (i.e., provided a response of “very difficult” or “difficult” to at least one of the three health literacy questions).

### 3.1. What Sources Were Highly Trusted and Frequently Used by Our Participants for COVID-19 Information?

#### 3.1.1. Trust

Among those who indicated their levels of difficulty in finding information, understanding oral information, and understanding written information (n = 676), the most trusted sources of COVID-19 information were doctors and other health care providers (M = 3.24, SD = 0.70), the CDC (M = 3.07, SD = 0.88), the WHO (M = 2.86, SD = 0.97), state/county/city health departments (M = 2.77, SD = 0.84), and official government websites (M = 2.56, SD = 0.91). The least trusted sources were social media (M = 1.60, SD = 0.66), (former) President Trump (M = 1.70, SD = 0.91), classmates (M = 1.71, SD = 0.59), TV (M = 1.76, SD = 0.72), and coworkers (M = 1.78, SD = 0.64).

We observed the same patterns among our participants who were at a high risk of having low health literacy (*n* = 150), as well as among those who chose either “I don’t look for health information” or “I don’t pay attention to written health information” (*n* = 87). However, we noticed that those who reported being at risk of having low health literacy had a lower levels of trust in various sources when compared to those who reported having adequate health literacy. Those who chose either “I don’t look for health information” or “I don’t pay attention to written health information” had the lowest levels of trust for various sources among these three groups.

#### 3.1.2. Use

Use of a source was positively associated with trust in that source across all the sixteen sources (all *p* < 0.001). Among those who indicated their levels of difficulty in finding information, understanding oral information, and understanding written information (n = 676), the most commonly used sources (used more than 50% of time) of COVID-19 information were the CDC (M = 4.33, SD = 1.80), as well as doctors and other health care providers (M = 4.19, SD = 1.67). State/county/city health departments (M = 3.66, SD = 1.78), the WHO (M = 3.63, SD = 1.94), official government websites (M = 3.40, SD = 1.84), and family members (M = 3.20, SD = 1.63) were also used more than 30% of the time. The least commonly used sources were the state governor (M = 1.93, SD = 1.39), magazines and newspapers (M = 1.95, SD = 1.26), (former) President Trump (M = 1.96, SD = 1.54), radio and podcasts (M = 2.00, SD = 1.31), and classmates (M = 2.00, SD = 1.19).

We observed the same patterns among our participants who reported being at a high risk of having low health literacy, as well as among those who chose either “I don’t look for health information” or “I don’t pay attention to written health information”. However, we noticed that those who reported being at risk of having low health literacy used various sources less frequently when compared to those who reported having adequate health literacy. Those who chose either “I don’t look for health information” or “I don’t pay attention to written health information” had the lowest frequency of using various sources among these three groups.

### 3.2. How Does Health Literacy Play a Role in Our Participants’ Trust in and Use of a Range of Potential Health Information Sources for COVID-19?

#### 3.2.1. Trust

Among those who reported their health literacy (*n* = 676), our unadjusted linear regression models indicated that students who reported having lower health literacy were more likely to trust COVID-19 information from social media (b = −0.04, *p* = 0.006) when compared to those who reported having higher health literacy. However, they were less likely to trust coworkers (b = 0.04, *p* = 0.022), (former) President Trump (b = 0.06, *p* = 0.010), and radio and podcasts (b = 0.04, *p* = 0.022).

As shown in Table 3, after controlling for demographic characteristics (i.e., gender, age, race, ethnicity, and income), students who reported having lower health literacy were also less likely to trust COVID-19 information from family members (b = 0.06, *p* = 0.005), coworkers (b = 0.04, *p* = 0.009), (former) President Trump (b = 0.05, *p* = 0.019), and radio and podcasts (b = 0.04, *p* = 0.018) when compared to those who reported having higher health literacy.

Our independent *t*-test results indicated that those who chose either “I don’t look for health information” or “I don’t pay attention to written health information” had significantly lower trust in doctors and other health care providers (*p* = 0.038), official government websites (*p* = 0.043), and the CDC (*p* = 0.006) for COVID-19 information when compared to those who reported their levels of difficulty in finding information, understanding oral information, and understanding written information.

#### 3.2.2. Use

Among those who reported their health literacy (*n* = 676), our unadjusted linear regression models indicated that students who reported having lower health literacy more frequently used social media (b = −0.13, *p* = 0.001) as sources of COVID-19 information than students who reported having higher health literacy. However, when compared to people who reported having higher health literacy, those who reported having lower health literacy less frequently used health care providers (b = 0.09, *p* = 0.027) as sources of health information about COVID-19.

As shown in Table 3, after controlling for demographic characteristics (i.e., gender, age, race, ethnicity, income), students who reported having lower health literacy more frequently used social media (b = −0.09, *p* = 0.032) as sources of COVID-19 information than students who reported having higher health literacy. However, those reporting a lower health literacy capacity were less likely to use health care providers (b = 0.12, *p* = 0.005), and the CDC (b = 0.11, *p* = 0.016) to obtain information about COVID-19 when compared to participants who reported having higher health literacy.

Our independent *t*-test results indicated that those who chose either “I don’t look for health information” or “I don’t pay attention to written health information” had a significantly lower frequency of using magazines and newspapers (*p* = 0.034), doctors and other health care providers (*p* = 0.002), official government websites (*p* = 0.002), the WHO (*p* = 0.031), the CDC (*p* = 0.018), and state/county/city health departments (*p* = 0.017) as sources of COVID-19 information when compared to those who reported their levels of difficulty in finding information, understanding oral information, and understanding written information.

## 4. Discussion

In our study of college students during the COVID-19 pandemic, we found that college students who chose either “I don’t look for health information” or “I don’t pay attention to written health information” had significantly lower trust in and use of official health authority sources (e.g., health care providers, official government websites, and the CDC) of COVID-19 information when compared to students who reported their levels of difficulty in finding information, understanding oral information, and understanding written information. Additionally, we also found that those who reported lower health literacy had a lower frequency of using official health authority sources (e.g., health care providers and the CDC) of COVID-19 information when compared to students who reported higher health literacy; however, students who reported lower health literacy used social media for COVID-19 information more frequently. This pattern of information-seeking behavior is similar to a previous study investigating the relationship between health literacy and use of and trust in sources for general health information, which found that lower health literacy was associated with lower odds of trusting in specialist doctors and dentists for health information, as well as using medical websites, but higher odds of using social media, among a U.S. nationally representative adult sample [13]. Similarly, a recent study conducted among a German-speaking adult population of Switzerland also found that those with lower health literacy tended to not trust in, and more rarely use, health professionals and health authorities as sources of COVID-19 information when compared to those with higher health literacy [24].

Several factors might help explain why people with low health literacy neither trust nor use health professionals and health authorities for COVID-19 information gathering. First, people with lower health literacy have difficulty understanding physician instructions and have negative perceptions of their healthcare experience, such as receiving inadequate health information, which contributes to a low trust in health care providers [13,25,26]. In fact, many people with low health literacy work hard to hide the fact that they have difficulty understanding oral or written instructions from health care providers; in addition, people with low health literacy often do not have a regular health care provider [27,28]. These factors create a challenge for health care providers in gaining trust among those with low health literacy and providing health recommendations.

The second possible explanation as to why people with low health literacy neither trust nor use health professionals and health authorities for COVID-19 information is that individuals with low health literacy have low trust in scientists, especially during the COVID-19 pandemic [29,30]. For example, people with low health literacy tend to not embrace the COVID-19 vaccine, due to a lack of trust in the government and scientists that stems from the uncertain attitudes towards possible herd immunity, the complexity of the scientific and political discourse surrounding COVID-19 vaccines, as well as questions about the vaccination effectiveness as new variants keep evolving [29]. Additionally, official rules and recommendations for preventive measures (e.g., mask mandates) based on scientific findings and clinical trials keep changing, because of the evolving knowledge about how this virus is behaving [30], which leads to negative feelings, such as being overwhelmed, confused, upset, and scared [8]. In fact, people with lower health literacy are more likely to avoid health information related to COVID-19, in order to reduce the above unwanted emotions [9].

Third, the level of trust in the CDC declined significantly during the COVID-19 pandemic, and many people view the CDC as strongly politized [31]. Another study also found that about 40 percent of adults in the U.S. felt that the CDC was paying too much attention to politics when issuing guidelines and recommendations for the COVID-19 policy [32]. Such distrust in the government stimulates the spread of fake news and mis/disinformation [33,34], which further reduces the trust in COVID-19 health information from government sources. Our current study finding indicates that college students with low health literacy have had an especially low trust in the government (e.g., the CDC) during this pandemic.

Several strategies can help to increase trust in health care providers and government agencies among people with low health literacy. First, it is critical to provide easy-to-understand health information, with plain, jargon-free language through various channels, in order to meet people where they are. Second, health care providers should use the teach-back technique with their low health literacy patients, in order to ensure these patients understand the information and instructions they have been given. When applying the teach-back technique, health care providers ask their patients to repeat, in their own words, what they have been told, using a caring tone of voice and attitude to create a “shame-free” environment for patients [35]. Third, collaborating closely with community organizations and social workers to create health messages with no political interference is another effective strategy for government agencies to utilize, in order to build trust in the public [29,36]. Another strategy for government agencies to maintain public trust is to ensure clear information and unambiguous health instructions related to COVID-19 that represent government transparency and effective communication [37]. Lastly, a prior study pointed out that the CDC should create COVID-19 health information with a clearer and more explicit focus on the science, and should provide rationales for any decision made regarding the guidelines and recommendations of COVID-19 policy [32].

Moreover, we found that college students who reported having low health literacy tended to use various sources (including health care providers and public health authorities, such as the CDC, the WHO, state/county/city health departments, and official government websites) less frequently, and trusted them less when compared to those who reported having an adequate health literacy capacity, which is consistent with a recent study [24]. This finding indicates that people with a higher health literacy might compare information among different sources, in order to check for trustworthiness, while those with a lower health literacy do not [24]. Furthermore, in one of our recent studies, we found that college students with lower health literacy were more likely to intentionally avoid information about COVID-19 [9]. Our findings confirm that college students with low health literacy are at risk of COVID-19 knowledge deficits, due to their high information avoidance. They might not learn about the most important preventive behaviors and the value of vaccinations, which could lead to compliance violations and vaccine hesitancy. Our findings also indicate that there is a critical need for higher education institutions to create learning outcomes and to provide training, in order to enhance health literacy skills among college students [38].

We found that health care providers, the CDC, the WHO, state/county/city health departments, and official government websites were frequently used and highly trusted sources of COVID-19 information among college students. This finding aligns with a previous study, which reported that the U.S. adult population used and trusted health care providers the most for general health information [13]. Another study also reported that doctors and government health agencies were the most trusted source, among a U.S. nationally representative adult sample, of general health information [39]. Social media had low trust among our participants, which is consistent with a recent study concluding that people had low trust towards social media as a source of COVID-19 information [24], as well with as a previous study, reporting that social media received low trust among the U.S. adult population as a source of general health information [13]. We also found that (former) President Trump was one of the least used and trusted sources for COVID-19 information. These results were similar to those reported in our recent qualitative study, which recruited participants from the same university, and found that college students identified various pieces COVID-19 misinformation, especially from social media and politicians [8].

However, COVID-19 related information-seeking behavior patterns might be different across countries. For example, a recent study conducted by De Gani’s team among 1012 participants (age mean = 46.2, SD = 16.9) living in the German-speaking part of Switzerland reported that television and the Internet were the most used information sources for COVID-19; health authorities and health professionals were used by less than half of the respondents, but these two sources were reported as being highly trusted [24]. Although we collected our data at the same time as De Gani’s team did, we found that television was not a frequently used source among our college participants, and they used health authority- and health professional-related sources frequently. These two study samples consisted of different age groups, however, which might contribute to such differences, as age is a significant predictor of information-seeking behaviors [40]. Another possible explanation is that COVID-19 information dissemination and mitigation strategies are used differently across countries [41].

Interestingly, after adjusting for gender, age, race, ethnicity, and income, the association between lower health literacy and higher trust in social media for COVID-19 information became non-significant. The 2007 HINTS data indicated that younger adults had a higher level of trust in online health information, regardless of the information quality; one possible explanation is that young adults perceive themselves as being less vulnerable to low-quality health information, due to this age group being generally healthier than older adults [42]. Therefore, our finding might also indicate that age plays a more critical role than health literacy levels in terms of trust in social media.

Social media is one of the least trusted sources for COVID-19 information among our participants; however, we found that students who reported having a lower health literacy more frequently used social media as sources of COVID-19 information than students who reported having higher health literacy. Social media can play a positive or negative role in providing health information related to COVID-19. Social media can be used to assist in seeking, understanding, and sharing health information; however, the quality and accuracy of the health information from social media need to be evaluated cautiously [43]. On one hand, social media helps to reduce social isolation and improve mental health, as it provides a connection between people and their peers, friends, and family [44]. On the other hand, COVID-19 misinformation is more likely to be shared on social media than official sources [45], and people experience negative feelings, such as fear and confusion, when they identify various pieces of misinformation related to COVID-19 on social media [8].

### Limitations

Although this study reported how health literacy plays a role in COVID-19 information-seeking behaviors, it has some limitations. First, the cross-sectional survey design of this study limits our ability to infer causal relationships. Second, different types of health literacy measures are associated with people’s patterns of information source usage [46]. There are many different instruments used to measure individuals’ health literacy [47]. Moreover, some studies have developed health literacy measures that are specific to COVID-19 [24,46]. Due to the fact that people have been constantly exposed to COVID-19-related information during this pandemic, they tend to have higher COVID-19-specific health literacy scores than their general health literacy scores [24]. Different measures of health literacy could produce different results, as different measures may assess slightly different skills [47]. Third, our convenience sampling method and relatively small sample size temper our ability to generalize our findings to the entire U.S. or other countries. Lastly, our COVID-19 information source list was not exhaustive, although the sources used here have been shown to be the most frequently used in many other studies.

## 5. Conclusions

This study makes an important contribution to our understanding of the patterns in information source preferences, specific to COVID-19, among college students attending a land-grant university located in the South–Central region of the U.S., with variations in self-reported health literacy levels. We found that college students who reported having lower health literacy reported lower levels of using and trusting official sources for COVID-19 information when compared to their peers who reported having higher health literacy. Relying on low-quality information sources and misinformation could create and reinforce people’s misperceptions regarding the virus, and further lead to less compliance with COVID-19-related public health measures.

## Figures and Tables

**Table 1 healthcare-11-00831-t001:** Outcome variables, questions, information sources, and response modes.

Variables	Questions Asked	Response	Information Sources (16 Items)
Levels of trust in information sources for COVID-19	How much do you trust the following sources to provide accurate coronavirus (COVID-19) information?	1 = Not at all 2 = Somewhat 3 = Mostly 4 = Completely	Social media (e.g., TikTok, Reddit, Instagram, Twitter) Magazines and newspapers Friends Family members Coworkers Classmates Doctors and other health care providers Official government websites President (previous) Trump State Governor City Mayor World Health Organization (WHO) Centers for Disease Control and Prevention (CDC) State, county, and city health departments TV (e.g., CNN, ABC, CBS) Radio and podcasts
Use sources for COVID-19 health information	In the past few months, how often have you used the following sources to get information about coronavirus (COVID-19)?	1 = Never 2 = Rarely, 10% of time 3 = Occasionally, 30% of time 4 = Sometimes, 50% of time 5 = Frequently, 70% of time 6 = Usually, 90% of time 7 = Every time	

**Table 2 healthcare-11-00831-t002:** Participants’ demographic characteristics (*n* = 763).

Demographics	*n*	%
Gender		
Male (including transmen)	248	32.50
Female (including transwomen)	506	66.32
Prefer to self-describe	9	1.18
Race		
White	564	73.92
Black	33	4.33
Native American	36	4.72
Native Hawaiian or Pacific Islander	1	0.13
Asian	21	2.75
Multiple Selected	99	12.98
Missing	9	1.18
Ethnicity		
Hispanic/Latino	59	7.73
Not Hispanic/Latino	704	92.27
Household income		
<$20,000	104	13.63
$20,000–$34,999	77	10.09
$35,000–$49,999	100	13.11
$50,000–$74,999	157	20.58
$75,000–$99,999	112	14.68
Over $100,000	203	26.61
Missing	10	1.31

Note. Due to rounding, some percentages do not sum to 100%.

**Table 3 healthcare-11-00831-t003:** Health literacy, and use of and trust in sources for COVID-19 information.

Sources	Trust			Use		
	b	95% CI	*p*	b	95% CI	*p*
**Social media**	−0.02	[−0.06, 0.01]	0.162	**−0.09**	**[−0.16, −0.01]**	**0.032**
Magazines and newspapers	−0.01	[−0.04, 0.02]	0.597	−0.02	[−0.08, 0.04]	0.540
Friends	0.00	[−0.03, 0.03]	0.836	−0.03	[−0.10, 0.04]	0.375
**Family members**	**0.05**	**[0.02, 0.09]**	**0.005**	0.06	[−0.02, 0.14]	0.125
**Coworkers**	**0.04**	**[0.01, 0.08]**	**0.009**	−0.01	[−0.08, 0.06]	0.777
Classmates	0.01	[−0.02, 0.04]	0.430	−0.03	[−0.09, 0.03]	0.378
**Health care providers**	0.03	[−0.00, 0.07]	0.061	**0.12**	**[0.04, 0.20]**	**0.005**
Official government websites	0.03	[−0.01, 0.08]	0.183	0.07	[−0.02, 0.17]	0.125
**President Trump**	**0.05**	**[0.01, 0.10]**	**0.019**	0.04	[−0.04, 0.12]	0.332
State Governor	0.03	[−0.01, 0.07]	0.088	0.00	[−0.07, 0.07]	0.969
City Mayor	0.01	[−0.03, 0.05]	0.708	0.01	[−0.06, 0.08]	0.792
WHO	0.01	[−0.04, 0.06]	0.754	0.07	[−0.03, 0.17]	0.157
**CDC**	0.04	[−0.00, 0.08]	0.079	**0.11**	**[0.02, 0.20]**	**0.016**
State/county/city health departments	0.03	[−0.01, 0.07]	0.147	0.07	[−0.02, 0.15]	0.144
TV	0.02	[−0.02, 0.05]	0.305	0.00	[−0.07, 0.08]	0.903
**Radio and podcasts**	**0.04**	**[0.01, 0.07]**	**0.018**	0.01	[−0.06, 0.07]	0.827

Note. Adjusted for gender, age, race, ethnicity, and income; bold font indicates statistically significant findings

## Data Availability

De-identified data may be made available upon reasonable request.

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
