# Peer review of "Trust in and Use of COVID-19 Information Sources Differs by Health Literacy among College Students"

_healthcare, 2023, doi:10.3390/healthcare11060831_

Round 1

Reviewer 1 Report

It is an interesting paper that stresses the linkages between health literacy and trusting official health authority sources for COVID-19 information. 

  • The low level of health literacy is a matter of unequal diffusion of information/knowledge. The authors should stress the importance of the diffusion of knowledge in achieving sustainable development goals. A piece that should be included here is Meramveliotakis and Manioudis, “History, Knowledge, and Sustainable Economic Development: The Contribution of John Stuart Mill’s Grand Stage Theory”, Sustainability, 2021, 13, 1468. 
  • Crisis management is critical in facing “exogenous shocks” such as COVID-19 and achieving sustainable development. An interesting piece here is Abbas et al., “The Role of Social Media in the Advent of COVID-19 Pandemic: Crisis Management, Mental Health Challenges and Implications”, Risk Management Healthc Policy, 2021, 14, 1917-1932. Have Social Media enhanced (or not) e-health literacy? 
  • It would be constructive to specify information sources (16 items) in table 1. For instance, do authors refer to specific magazines and newspapers? What are the official government websites?  
  • It would be constructive for the paper to add some information regarding data analysis (section 2.3). 
  • Is health literacy associated with income? This correlation should be more explicit in the discussion section.

Specific points:

  • Generally, a 5-point or a 7-point scale is preferable to a 4-point scale.
  • Please add the word “type” between Likert and scale in line 119.

Author Response

We appreciate the reviewer's comments. The comments were very helpful as we revised sections of our paper and we believe the changes we made have resulted in a stronger contribution to the literature. In the attachment, we address each comment in turn, and indicate where in the manuscript text to find specific revisions and newly added information. 

Reviewer 2 Report

The manuscript examines the relationship between health literacy and people’s trust in and use of a range of potential health information sources for COVID-19 among college students. Paper is well written.

Some major comments.

The variable selection method applied in the multivariable model is unclear.

Table 3 is unclear. Which are the outcomes and which the predictors?

The actual impact of adopting preventive measures is lacking. For example, this recent systematic review did not find any strong association between health literacy and vaccination status (Siena LM, et al. The Association of Health Literacy with Intention to Vaccinate and Vaccination Status: A Systematic Review. Vaccines (Basel). 2022 Oct 29;10(11):1832.). Please elaborate on it.

Author Response

(The authors gave the same response as above.)

Reviewer 3 Report

I consider that the article, based on the elements presented, is a good proposal. The recommendations offered are the following:
a). Review the writing, especially in the introduction, since it has very long paragraphs that include several ideas together, it is necessary to separate them since otherwise they leave the reader breathless.
b). They must complement the keywords, it is necessary to consult a Thesaurus or other scientific articles already published on the subject, for example: trust and use do not represent anything for a search.
c). It is recommended to add a brief theoretical framework between introduction-materials and methods where evidence from other research in this regard is presented and key concepts such as Health Literacy are defined, or at least what is understood in this article.
d). In the case of a research developed in the USA and its racial variety, the classification used is not clear, they should use the racial classification of the US census. You divide racial and ethnic groups without any need, in ethnic terms they speak of Hispanics, Latinos and people of Spanish origin and in the tables I think they use Hispanic/Latino which is the most correct; also state that the income of the participants is expressed in US dollars.

Author Response

(The authors gave the same response as above.)

Round 2

Reviewer 1 Report

I believe that the authors have sufficiently addressed my previous comments and that their additions have improved the paper.

Reviewer 2 Report

--

Reviewer 3 Report

The authors satisfactorily addressed the observations made in the previous evaluation. I consider that the article is fit for publication.